# Could Primary Chemoradiotherapy in T2 Glottic Cancers Yield Results Comparable to Primary Radiotherapy in T1? Considerations from 531 German Early Stage Patients

**DOI:** 10.3390/cancers13071601

**Published:** 2021-03-31

**Authors:** Gerhard Dyckhoff, Rolf Warta, Christel Herold-Mende, Elisabeth Rudolph, Peter K. Plinkert, Heribert Ramroth

**Affiliations:** 1Department of Otorhinolaryngology, Head and Neck Surgery, University of Heidelberg, 69120 Heidelberg, Germany; Rolf.Warta@med.uni-heidelberg.de (R.W.); Christel.Herold-Mende@med.uni-heidelberg.de (C.H.-M.); Peter.Plinkert@med.uni-heidelberg.de (P.K.P.); 2Division of Neurosurgical Research, Department of Neurosurgery, University of Heidelberg, 69120 Heidelberg, Germany; 3Heidelberg Institute of Global Health, University of Heidelberg, 69120 Heidelberg, Germany; rudolph_medizin@web.de (E.R.); Heribert.Ramroth@uni-heidelberg.de (H.R.)

**Keywords:** laryngeal cancer, organ preservation, radiotherapy, radiochemotherapy, transoral laser microsurgery, survival

## Abstract

**Simple Summary:**

T1 laryngeal carcinoma arising from the vocal cords (glottis) is a highly curable disease with local control (LC) rates of over 90% by either primary radiotherapy (pRT) or transoral laser microsurgery (TLM). For slightly larger glottic tumors (T2), the outcome is significantly poorer. However, in the case of recurrent tumor after pRT, to save the patient’s life, the larynx has often to be removed (total laryngectomy). A remedy could bring more effective radiotherapy. In a large observational study on laryngeal cancer, a small number of early-stage patients received chemotherapy in addition to primary radiotherapy (pCRT). After pCRT, more patients could be saved from a recurrent tumor, lived markedly longer, and could preserve their functional larynx. pCRT can cause more side effects. However, according to the literature, for early-stage laryngeal cancer, they should be well controllable. To prove the increased effectiveness and acceptable toxicity, studies with more patients need to be conducted.

**Abstract:**

T1 glottic cancer is a highly treatable disease with local control (LC) rates over 90% by either primary radiotherapy (pRT) or transoral laser microsurgery (TLM). LC of T2 glottic cancers is 15 percent points poorer on average. However, salvage after pRT entails more than 50% total laryngectomy. Therefore, there is a need for enhanced LC. Altered fractionation regimens improved LC in T1 but not in T2. For this reason, for T2, alternative strategies must be considered. In a large observational cohort study including 531 early-stage laryngeal cancers, a small number of patients were treated with primary chemoradiotherapy (pCRT). In multivariable analysis, factors associated with significantly poorer outcomes included age, comorbidities, supraglottic localization, and T category. While there was a significant difference between pRT and surgery (HR 1.79; 95%-CI: 1.15–2.79), there was none between pCRT and surgery (HR 0.70; 95%-CI: 0.33–1.51). There is evidence from the literature that pCRT in early glottic cancers could yield results that surpass the limits so far experienced in radiotherapy alone with acceptable toxicity. Thus, prospective randomized studies with larger numbers of patients are warranted.

## 1. Introduction

Laryngeal carcinoma is the most frequent malignancy in otolaryngology [1]. In 2018, the worldwide estimate of new cases was 177,422, and the number of deaths caused by laryngeal cancer was 94,771 [2]. The age-standardized incidence rate was more than 7 times higher in men than in women [1]. As consistently shown, the main risk factors for laryngeal cancer are smoking and alcohol consumption [3,4,5]. Occupational exposure to asbestos, polycyclic aromatic hydrocarbons, and cement dust plays a comparatively minor role [3,6,7,8,9]. Similarly, HPV infection seems to have a causative role only in a small subgroup of laryngeal cancers [10].

The success of tumor treatment is a question of not only overall survival (OS) but also functionality and quality of life. This is especially true for laryngeal cancer because of the crucial roles the larynx plays in speaking and swallowing. In the early stage, for an entire century, pRT was a key component in the management of laryngeal malignancies with open partial laryngectomy (OPL) as a surgical alternative [11]. Starting in 1972 [12], transoral laser microsurgery (TLM) was stepwise introduced as a new, less invasive treatment option with excellent functional outcomes and without the necessity of temporary tracheostomy, which is usually mandatory in OPL [13]. To date, according to the guidelines of the National Comprehensive Cancer Network (NCCN), the treatment of choice for early-stage laryngeal cancer is single-modality therapy with either larynx-preserving surgery alone (generally TLM) or definitive radiotherapy alone [14].

For the advanced stage, almost thirty years ago, larynx-preservation multimodality protocols were introduced. Two large randomized prospective studies have been conducted, yielding level I evidence that larynx-preservation by pCRT is a treatment option comparable to laryngectomy followed by aRT “without jeopardizing survival” [15,16]. For the early stage, however, thus far, there have been no head-to-head comparative randomized prospective studies. Thus, there is no compelling evidence favoring one technique over the other in terms of oncological and/or functional outcomes [17]. From recent meta-analyses, however, there is growing evidence, that although pRT and TLM are equivalent in terms of local control (LC), there may be significant differences in overall survival (OS) and laryngeal preservation (LP) [18,19,20,21]. Early glottic carcinoma (Tis-T2) is regarded as a highly treatable disease [22]. While this is undoubtedly true for Tis-T1 with LC rates of more than 90% after TLM and pRT, the outcome of T2 is significantly less favorable [23,24,25,26,27,28,29]. In the case of recurrent tumor after pRT, patients often have to undergo total laryngectomy [13,30]. To prevent salvage laryngectomy, local control rates should be improved. By altered fractionation, the outcome of T1 glottic carcinomas after pRT could be significantly increased in terms of LC [30]. However, this benefit did not persist for T2 tumors, so “alternative strategies should be considered” [30]. In the evaluation of our large observational cohort study on incident laryngeal cancer patients, we made some intriguing observations in the early stage, which offers the opportunity to develop a potent alternative strategy.

## 2. Materials and Methods

As reported previously [31,32], we recruited all index laryngeal cancer patients from the five clinical centers in the Rhein-Neckar-Odenwald region in southwest Germany who were in charge of the treatment of laryngeal cancer patients between 5 January 1998 and 31 December 2004. Patients for this study were identified in two different ways. The first part of the patient cohort was patients who participated in a previous prospective case-control study between 1998 and 2000. For the second part (2001–2004), all patients were identified retrospectively. The charts of all patients were evaluated retrospectively. Demographic data and clinical information were extracted from hospital medical records using a standardized form. Patients were followed up until March 2015 to capture potential recurrences (including rTNM), second primaries, and their treatment. Vital status and date and cause of death were requested from local registries. Overall survival rates were calculated using the Kaplan-Meier (KM) method. Regression analysis was performed using multivariable proportional hazards models adjusted for the confounders sex, age, comorbidities, TNM stage, differentiation, and primary tumor site. Only overall survival (OS) and disease-specific survival (DSS) estimates are presented. The OS rates of pCRT and pRT, both with the option of salvage total laryngectomy, were compared with those of surgery with adjuvant radiotherapy or adjuvant chemoradiotherapy, as indicated by risk and stage (surgery ± a(C)RT). Survival time was measured as the time from the first diagnosis until death or until 21 March 2015. To describe treatment success in terms of larynx preservation, two different variables were used: (1) larynx preservation time (LP), i.e., the time until laryngectomy. In this case, only laryngectomy was regarded as an event. (2) laryngectomy-free survival time (LFS), i.e., overall survival time with preserved larynx—in this case, laryngectomy and death were regarded as events. For the evaluation, patients who moved away from Germany were censored one month after emigration. *p*-values below 0.05 were regarded as statistically significant. The following variables showed an effect in the univariate analysis (*p* < 0.20) and were included in the multivariable analysis as explanatory variables: age at first diagnosis (continuous), comorbidities, tumor site, TNM status, and therapy modality. Backward selection was used to receive a final model. The proportional hazards assumption was checked by adding a time-dependent version of all the variables in the model [33]. The assumption was met for all variables. The metastatic status could not be evaluated as M1 status could be clearly determined for only 5 patients. Comorbidity conditions were determined using the Charlson Comorbidity Index (CCI), which summarizes 18 different comorbidities, weighted by severity, in a single score [34]. For this analysis, we considered the binary form of the variable, which is set to one for CCI values of two or higher. The data analysis for this paper was generated using SAS/STAT software, version 14.2, of the SAS System for Windows, copyright © 2012 (SAS Institute Inc., Cary, NC, USA).

## 3. Results

The study region covers a population of approximately 2.7 million people. In total, the study cohort consisted of 810 index laryngeal cancer patients. Fifty-three patients were excluded from evaluation because they either received no treatment with curative intent (*n* = 28), had an unknown tumor stage (*n* = 13), or presented with synchronous second primary disease at first diagnosis (*n* = 12), ultimately resulting in a cohort of 757 patients. A total of 531 patients had early-stage laryngeal carcinoma (T1 and T2). Within the surgical therapy group (*n* = 674; 89.0%), most of the patients were treated by transoral laser microsurgery (TLM) (*n* = 443; 65.7%), a minor portion received open partial laryngectomy (OPL) (*n* = 59; 8.8%), and in 172 patients (25.5%) a total laryngectomy (TL) was performed. According to stage and resection status, 143 patients (21.2%) received adjuvant radiotherapy (aRT), and 22 patients got adjuvant chemoradiotherapy (aCRT) (3.3%). Eighty-three patients (11.0%) received primary conservative treatment: 38 patients (5.0%) were treated by platinum-based pCRT and 45 patients (5.9%) received pRT alone. Table 1 shows the demographic and clinical overview of the laryngeal cancer patients of the cohort within the three treatment arms.

Table 2 gives an overview of the 5- and 10-year overall and disease specific survival rates in the different treatment arms over T category.

Age and Charlson Comorbidity Index (CCI): In the T1 group, the patients who received surgery ranged from 34 to 91 years, and the median age was 63.4 years. The median in pRT patients was slightly higher (65.3 years), while the median in patients receiving pCRT was markedly lower (54.8 years). In the T2 group, patients undergoing surgery and pCRT were approximately the same age (61 and 60 years), while the pRT patients were considerably older (68.5 years). A total of 71.7% of patients in the overall cohort had no comorbidities, (i.e., a CCI = 0). The percentage of pCRT patients with a score equal to zero was slightly higher (81.6%), whereas for pRT patients the percentage was lower (48.9%). The latter were based on smaller numbers only.

In a multivariate Cox regression comparing T2 to T1 laryngeal cancer patients including known confounders such as age, sex, CCI, primary tumor site, tumor category and therapy, the following factors were found to be statistically significant: age (hazard ratio [HR] 1.57; 95%-(confidence interval) CI: 1.37–1.81), comorbidities (HR 1.56; 95%-CI: 1.21–2.02), and supraglottic localization of primary tumors (HR 1.93; 95%-CI: 1.43–2.59). For tumor category T2 vs. T1, there was a strong tendency without reaching the level of significance (HR 1.28; 95%-CI: 0.98–1.66). While there was a significant difference between pRT and surgery (HR 1.79; 95%-CI: 1.15–2.79), there was no significant difference between pCRT and surgery (HR 0.70; 95%-CI: 0.33–1.51). (Table 3).

In tumor category T2, there were 18 and 9 patients treated by pRT and pCRT, respectively, compared to 173 patients who received primary surgery. Kaplan-Meier curves were constructed to illustrate the different outcomes descriptively (Figure 1).

The treatment outcomes after TLM for early laryngeal cancer (T1, T2, and T1 + T2) after 5 and 10 years are given in Table 4 in terms of OS and DSS. For larynx preservation, data are given as the larynx preservation rate (LP) and the laryngectomy-free survival rate (LFS).

None of the T1 patients in the pCRT group (0/5) experienced total laryngectomy (TL), but there was one TL in the pRT group (1/12). For the T2 patients, there was 1 TL in the pCRT group (1/9) and 4 in the pRT group (4/18). A multivariable Cox regression analysis of all early-stage patients treated with the primary intention of larynx preservation (pRT: *n* = 30, pCRT: *n* = 14, TLM: *n* = 399) revealed a significantly higher probability of undergoing TL after pRT than after TLM (HR 3.31; 95%-CI: 1.28–8.53). No significant difference could be seen between pCRT and TLM (HR 0.67; 95%-CI: 0.09–4.87).

## 4. Discussion

Our observational cohort study on 757 incident laryngeal tumor patients revealed that nonsurgical treatment of laryngeal cancer was already an option but not a standard treatment in Germany 20 years ago. In all tumor categories, the main treatment approach was surgery: T1: 94.9%, T2: 86.5%, T3: 85.1%, and T4: 80.0%. Eighty-nine percent of the overall patient cohort was treated surgically, and only 5.0% and 5.9% were treated with pCRT and pRT, respectively. This is important to note, as even in neighboring countries such as Denmark, the treatment in the same time period was almost exclusively conservative. The unique data published by the DAHANCA group evaluating the nationwide outcomes of all Danish glottic cancer patients between 1971 and 2011 reported nonsurgical treatment in 98% of patients [35]. In their cohort, including all T categories, the rate of pCRT was only 2%. In our cohort recruited between 1998 and 2004, within the small conservative treatment group of only 11% of patients (*n* = 83), an increased proportion of CRT with advanced T category could be seen: T1: 29.4%, T2: 33.3%, T3: 68.8, and T4: 61.9%.

### 4.1. Could pCRT in T2 Be Superior to Surgery in T1?

In univariable analyses, in the T1 group, the 5-year DSS rates after pRT and pCRT were higher than those in patients after surgical treatment (100% after pRT and pCRT; 95% after surgery) (Table 2). In the T2 group, there were only small numbers in the conservative arms (pCRT: *n* = 9 and pRT: *n* = 18 compared to surgery: *n* = 173). Univariately, the outcomes of T2 patients after pCRT were better than the outcomes of T1 patients after surgery (5- and 10-year OS: 89% and 67% for T2 after pCRT vs. 80% and 59% for T1 after surgery), some based on small numbers only. A multivariable Cox regression model was performed including the confounders age and CCI. While there was no significant difference between pCRT and surgery (HR 0.70; 95%-CI: 0.33–1.51), there was, however, a significant difference between pRT and surgery (HR 1.79; 95%-CI: 1.15–2.79).

### 4.2. After pRT and TLM: No Differences in LC, but TLM Better in OS 

According to the guidelines of the National Comprehensive Cancer Network (NCCN), the treatment of choice for early-stage laryngeal cancer is single-modality therapy with either larynx-preserving surgery alone (endoscopic resection or open partial laryngectomy) or definitive radiotherapy alone [14]. Thus, in the literature, the benchmark for the outcome of pRT of laryngeal cancer is generally TLM. In the advanced stage, there were two large prospective randomized studies comparing primary surgical and primary nonsurgical treatment strategies head-to-head [15,16]. However, for the early stage, studies of this type have not yet been conducted. Depending on the endpoints and the selection of studies included, meta-analyses and reviews come to different conclusions as to whether oncologic and functional outcomes are equivalent between these two treatment modalities for early glottic cancers [17]. Previous meta-analyses including only a few studies and predominantly studies from the beginning of laser surgery have only limited significance today. Feng retrieved only 3 studies [36,37,38] consisting of 317 patients (of which 131 were treated by TLM) between 1990 and 2000, which were analyzed for 5-year oncological outcomes. In his meta-analysis, no significant difference in LC after TLM and pRT was observed but no data on OS or DSS were reported [39]. Abdurehim observed significant heterogeneity in studies conducted before 2000 and after 2000. Depending on the source and dose of radiation, TLM was superior to pRT in one subgroup and vice versa in the other subgroup. Furthermore, in older studies, oncological outcomes after TLM were poorer than in those in more recent studies. The overall pooled effect showed that there was no significant difference with respect to LC and OS [40]. In the first large meta-analysis conducted by Higgins including over 7600 patients in 26 studies, there was also no significant difference in LC. However, TLM was significantly superior to pRT in terms of 5-year OS (HR 1.48; 95% CI: 1.19–1.85). Consistently, all recent meta-analyses showed no significant difference in LC, but there was a significant difference in OS [18,19,20,21] (Table 5).

As previous studies were dominated by Tis and T1 patients, Warner focused on T2 patients [43]. Consistent with the meta-analyses in Table 5, in his meta-analysis on more than 4300 T2 patients (TLM: *n* = 1.156 vs. pRT: *n* = 3.191, treated in 48 studies), no significant differences in LC were found. The weighted average 5-year LC rates were 75.8% after pRT and 77.3% after TLM [43]. OS and DSS were not considered in this study.

### 4.3. T2 Compared to T1: Significantly Poorer Outcomes after TLM and pRT

Early glottic carcinoma (Tis-T2) is regarded as a highly treatable disease [22]. While this is undoubtedly true for Tis-T1, the outcome for T2 is significantly less favorable. This is consistently shown in several large series of pRT studies in early glottic cancer (presented studies included at least 200 T2 patients and present data for T2 and T1) (Table 6).

In the majority of studies, for T1 stage a local control rate of more than 90% was achieved, at least in the T1a cases [23,25,27,28,29]. The local control rates for T2 tumors ranged between 63% and 79% for the overall group [23,24,26,27,28,29]. The T2a patients in one study reached 80% while in the same series the T2b patients showed an LC rate of 70% only [27]. On average, T2 patients scored approximately 15 percentage points worse than T1 patients (63–79 compared to 82–94). The difference between T1a and T2b amounted to even more than 20 percentage points [23,27]. In line with these findings, in the nationwide DAHANCA study which reported the outcomes of 1453 T2 patients compared to 2742 T1 patients (96% treated by pRT), the failure rate from T1 to T2 increased from 16.4% to 33.0% (HR 2.0; 95%-CI: 1.8–2.4) [35].

Considering TLM patients only, our study showed a similarly significant difference in outcome between T1 and T2 (HR 2.63; 95%-CI: 1.41–4.93). For T1 after TLM, similar to T1 after pRT, in the studies included in the meta-analyses of Table 4, LC rates of more than 90% were reported [44,45]. Thus, if there was a chance to enhance the effectiveness of conservative treatment in such a way that T2 laryngeal cancers could achieve results that even surpass those of T1 tumors after surgery, it would be worth investigating.

### 4.4. In T2: Can the Effectiveness of RT Alone Be Further Improved?

Over the course of an entire century, considerable improvements have been achieved in the efficacy of radiotherapy. After pioneering work was performed by Regaud and Coutard in the 1920s, radiotherapy became an important mainstay of tumor therapy [11]. In “conventional radiation”, the laryngeal fields consisted of two opposing lateral fields. By increasing the field from 5 × 5 cm^2^ to 6 × 6 cm^2^, Harwood reduced the recurrence rate in T1 and T2 by 9 percentage points in the early 1970s [46]. Today, the laryngeal field is defined by the superior border at the top of the thyroid cartilage and the inferior border at the bottom of the cricoid cartilage, for T2 one tracheal ring below the cricoid [47]. In the 1980s, the dose applied was increased from 1.8 to 2.0 Gy per fraction thus increasing local control rates in T2 glottic cancer from 16.7% to 74.1% [48]. By reducing the overall treatment time by altered fractionation, the LC could be further improved. By means of hypofractionation (increasing the dose per fraction) or hyperfractionation (increasing the number of fractions per day), local failure events could be reduced by 38% and 60%, respectively. However, this benefit was only found in T1 laryngeal carcinoma but did not persist in T2 tumor patients [30]. For T2 laryngeal carcinomas, the Radiation Therapy Oncology Group (RTOG) designed a large prospective study involving 250 patients that was empowered to detect an improvement in LC from 70% to 85% (RTOG 95-12 trial) [49]. By hyerfractionation (HFX) compared to standard fractionation (SFX), LC could be improved by 8 percentage points to 78%, though this improvement was not statistically significant and the set target was not achieved. The 5-year OS for T2 patients was 72% after HFX compared to 63% after SFX. The intensification of therapy resulted in an increase in acute grade 3+ toxicity from 22.7% to 33.3%, while the 5-year cumulative incidence of grade 3-4 late toxicities was 8.5% in both treatment arms. The question is whether in T2 glottic carcinomas, by altered fractionation, further significant improvements in LC and OS is possible or whether the limits of efficacy of pRT alone has been reached [30].

### 4.5. After pRT: Significantly Poorer LP

There is a direct correlation between local control and a parameter that has paramount importance for laryngeal cancer patients: larynx preservation. As Sapienza pointed out, the success of organ preservation directly depends on local control of the tumor. For ultimate local failure, surgical salvage entails total laryngectomy in more than half of cases [30]. Only in selected cases is organ preserving-surgery by TLM or OPL a possible procedure for salvage of failure after nonsurgical treatment [13]. The consequence of local recurrence for laryngeal preservation can exemplarily be seen in the study of Schrijvers et al. [38]. Nicely balanced study collectives were compared: 49 T1a patients were treated with TLM, and 51 T1a patients were treated with pRT. There were 13 recurrences in the TLM arm (27%) and 12 recurrences in the pRT arm (24%), resulting in a 5-year LC rate of 73% in the pRT group and 71% in the TLM group, thus there was no statistical difference in LC (*p* = 0.267). There was only one disease-specific death in the pRT group, and none in the TLM group, thus there was no significant difference in DSS. The decisive difference in outcome between both treatment approaches can be seen in the treatment of recurrent tumor patients. In the TLM group, 4 patients (31%) received successful TLM reresection, and the other 9 (69%) healed by radiotherapy. However, in the pRT arm, only one recurrent patient could be resected successfully by laser surgery, 9 patients (75%) had to undergo total laryngectomy and two received only palliative treatment. Thus, the 5-year laryngeal preservation rate was significantly higher in the TLM group than in the pRT group (95% vs. 77%, *p* = 0.04) [38]. These outcomes suggest a paradigm shift. We must not focus too much on the success rates of different therapy approaches but need to pay more attention to their failure rates. Decisive are not the 80–85% of successful local control rates which may be similar for TLM and pRT. Decisive are the 15 or 20% of primary treatment failures and their respective treatment options. After TLM, a whole set of therapeutic options is available: TLM reresection, OPL, and RT. After pRT, reradiation is normally not expedient and options for larynx sparing surgery are usually limited. However, the study of Schrijvers is not an isolated case. Consistently, all recent meta-analyses report LC rates without significant difference between TLM and pRT but highly significant differences in LP (Table 4). The only exception is Higgins’ report. In his meta-analysis, 26 studies were evaluated in terms of oncological outcomes. In terms of laryngectomy-free survival, however, only 4 studies were evaluated. In these four studies, the Spector study [29] had a weight of 51.8%. This study, however, was not representative for TLM studies in general. Published in 1999 the study reported rather preliminary results from the built-up phase of laser surgery. The surgical group was recruited between 1971 and 1990. A total of 404 of the 465 surgically treated T1 N0 M0 glottic cancer patients underwent open partial laryngectomy (OPL) and only 61 patients were treated by TLM. Today, OPL is performed in T1 glottic cancer only in exceptional cases. Six patients (9.8%) among the 61 TLM patients and 29/194 pRT (14.9%) underwent total laryngectomy [42]. The fact that in the built-up phase of TLM, results were poorer than after complete establishment of the method can be exemplarily seen in the Erlangen-Nuremberg center of excellence. Here, in 1998, almost at the same time as Spector, Iro published the first series of 141 TLM patients with an overall R0-resection rate of 78% [50]. After proper establishment of the technique, in 2011, Iro reported a rate of 94.8% in 557 glottic and 91.2% in 137 supraglottic T1 and T2 cancers [51]. The same effect of poor results in the built-up phase of TLM is explicitly described in the meta-analysis of Abdurehim et al. [40]. Among the 8 studies reporting data on larynx preservation, 3 were published before 2000 and 5 after 2000. Because of heterogeneity, these two groups were evaluated separately. In the before-2000 subgroup there was no significant difference in larynx preservation, while in the after-2000 subgroup the pooled OR of LP was 8.23 (95%-CI: 3.61–18.76) favoring TLS (*p* = 0.00001). In all recent meta-analyses reporting larynx preservation data from studies after 2000 [18,19,20,21], the HRs ranged between 3.85 (95%-CI: 1.92–7.72) [20] and 6.31 (95% CI: 3.77–10.56) [21] (Table 5). In line with these findings, in Hendriksma´s review on T2 glottic carcinomas, including studies between 2000 and 2017, a higher larynx preservation rate for patients treated primarily with TLM compared to pRT was reported (88.8 vs. 79.9%) [22]. In the DAHANCA study of all Danish laryngeal cancer patients treated between 1971 and 2011, Lyhne reported a 5-year laryngectomy-free survival (LFS) rate of 48% for 1453 T2 cancers. Nota bene, here the LFS was reported, i.e., OS with preserved larynx, not the larynx preservation rate, as in many other studies. In LP, an event is defined by laryngectomy only; in LFS, an event is defined by TL or death; thus, when considering LP only, a larynx may be deemed successfully preserved, although the patient is dead. In our retrospective cohort study, for T2 patients, the 5-year LP rate after TLM was 82% (95%-CI: 73–88) but the LFS rate was 65% (95%-CI: 56–73) (Table 4). As Argiris and Lefebvre pointed out, the LFS is a more comprehensive parameter than only “larynx in place (i.e., no laryngectomy)”, “as survival is an important issue” [52]. Even more comprehensive is laryngoesophageal dysfunction-free survival which was defined in a consensus panel for the design of prospective studies in 2009 [53]. However, in retrospective studies, this parameter is not applicable. Here, the LFS may be regarded as acceptable standard. Although statistically univariate data of different cohorts are not directly comparable due to different distributions of confounding variables, there is at least a tendency to poorer LFS when the conservatively treated T2 patients of Denmark are compared to the TLM-treated T2 patients of our study region in southwest Germany: LFS of 48% after pRT vs. 65% after TLM.

### 4.6. After pCRT: LP Rates Comparable to Those after Surgery

In our study, the 5-year LP rates after TLM were 93% for T1 and 82% for T2. Compared to the 5-year LP rates of glottic T2 cancers of 88.8% after TLM reported by Hendriksma, our univariable results appear quite poor. However, in addition to incomparability of univariable results, we have to take into consideration that in our TLM cohort, there were 34.2% supraglottic, transglottic and subglottic carcinomas. These are known to have a worse prognosis compared to glottic cancers. In a multivariable Cox regression model, in our study, supraglottic cancers had a significantly poorer prognosis than glottic carcinomas (HR 1.93; 95%-CI: 1.43–2.59, Table 3).

In a multivariable Cox regression model, we compared the outcome after nonsurgical treatment with that after TLM. After pRT, there was a significantly higher risk of undergoing TL than after TLM (HR 3.31; 95%-CI: 1.28–8.53). This HR is in line with the results shown in the meta-analyses in Table 5. However, there was no significant difference between pCRT and TLM (HR 0.67; 95%-CI: 0.09–4.87). Thus, in our study, pCRT was superior to pRT in terms of not only oncological outcomes but also larynx preservation.

### 4.7. pCRT for T2 Glottic Cancer in the Literature

After these favorable findings for pCRT in the T2 patients of our cohort study in terms of OS and LP, we conducted a literature search and found that the idea of pCRT in T2 laryngeal carcinomas is not new. For T2 tumors with impaired vocal fold mobility (formerly called T2b tumors), a poor prognosis was reported [54], referring to laser surgery [55,56], and pRT [24,57]. For these patients chemoradiation was considered [58] and recommended [59]. In a very recent communication, Hamauchi reports the outcomes after pCRT (*n* = 14) compared to pRT (*n* = 25) in T2 laryngeal cancers with high-risk factors (subglottic extension, impaired cord mobility, or bulky tumor size) [60]. 5-year locoregional control and survival rates after pCRT were better than after pRT (85.7% vs. 44.2%). In a multivariable analysis, CRT was associated with favorable oncological outcomes (*p* = 0.04) [60].

For T2 glottic laryngeal cancer patients in general, i.e., without special regard to a high-risk situation, Furusaka published three long-term series comparing pRT alone with pCRT either with a combination of cisplatin and 5-FU or two different concentrations of carboplatin (AUC 1.5 or AUC 2.0) [61,62,63]. The 5- and 10-year OS and LP rates are given in Table 7. In the high-dose carboplatin arm and in the cisplatin/5-FU combination arm, 5- and 10-year OS rates ranged over 95% and over 90% compared to 88.5% and 73.5% in the pRT alone arm. The 5- and 10-year LP were over 75% and over 73% in the pCRT arms and 60.4% and 50.1% in the pRT alone arm. The differences in outcome did not reach statistical significance because of low numbers of patients in each arm (between 25 and 57 patients).

In a recent study of Al Feghali et al. on T2 glottic cancers, 20 patients treated with pCRT were included [64]. Formally, no significant difference in oncological outcome after pCRT was reported. However, there was no separate evaluation of the different types of treatment intensification used, which consisted of addition of chemotherapy and/or altered fractionation. Furthermore, 49% of T2b patients received intensified therapy, while only 37% of T2a patients got some sort of intensification. Thus, the fact that there was no significant difference in outcome between T2b and T2a is an indicator that the intensification was effective. Consistently, the authors recommend treatment intensification for patients with impaired vocal cord mobility (T2b) as their new standard approach [64]. Comparing pRT + chemotherapy (pCRT or pRT + aCT) to pRT alone, Kitani et al. reported a significant lower incidence of distant metastases and secondary primaries (5-year incidence 5% vs. 19%, *p* < 0.05) and fewer deaths of theses causes (1 vs. 8, *p* < 0.05) in T2 laryngeal cancer patients. Interestingly, as chemotherapeutic agent he used S-1, a prodrug of 5-FU administered orally [65].

### 4.8. Concerns of Toxicity

Concerns have been raised against pCRT because of acute and particularly late toxicities. The total cumulative dose received by the surrounding normal tissue limits conventional radiation [11]. This radiation-induced toxicity is enhanced by the addition of chemotherapy [13]. In the radiation of early-stage laryngeal carcinoma, the salivary glands with resulting xerostomia and/or sticky saliva are not the main organs at risk. However, the thyroid glands (->hypothyroidism), the carotid arteries (-> cerebrovascular events), and particularly the inferior pharyngeal constrictor muscles (PCMs) with resulting dysphagia and/or aspiration are major concerns after p(C)RT of laryngeal malignancies [11].

As an example, Rudat et al. [66] conducted a study with pCRT in moderately advanced laryngeal and hypopharyngeal carcinoma (accelerated RT: 66 Gy in 5 weeks with concomitant boost technique, administered by opposing lateral fields +70 mg/m^2^ carboplatin) [66]. Nine of 42 patients (21%) exceeded the scheduled treatment time by ≥5 days because of acute toxicity, especially mucositis (41.6% grade 3+). Furthermore, 6 of 23 (26%) tumor-free long-term survivors received tracheotomy because of late laryngeal toxicity with dysphagia because of radiation-induced fibrosis 30–79 months after the completion of pCRT. Thus, the potential LP rate with the functional larynx was reduced from 81% to 67%. However, the authors speculate that the high rate of laryngeal toxicity was due to the concomitant boost fractionation regimen rather than to the addition of carboplatin. In the concurrent pCRT arm of the RTOG 21-11 study published by Forastière, standard fractionation was applied, and although cisplatin was added, an LP of 83.6% was reached [67]. Interestingly, in the RTOG 21-11 study, grade 3 or 4 toxic effects were observed in the concurrent pCRT arm in 30% and in the radiotherapy alone arm in 36% [68]. Thus, in this study, the addition of chemotherapy did not seem to have caused a higher incidence of late toxicities.

The studies of Rudat and Forastière were trials with moderately advanced tumors, the surgical treatment of which would have been TL. More relevant for the issue of toxicity of pCRT in T2 glottic cancers are the studies of Furusaka. The systemic 3+ side effects in the combination pCRT group were leukopenia and neutropenia in 2 (6.3%) patients and local 3+ side effects were mucositis in 11 (34.4%) and dermatitis in 9 (28.1%) patients [63]. All of these events were reportedly reversible. However, in the pRT alone group there were no systemic 3+ toxicities at all, and only 4 patients had grade 3 stomatitis [62]. The incidence of local 3+ side effects may be considered to correlate with the 3+ side effect rates in the RTOG 95-12 trial (SFX vs. HFX, without chemo) reported by Trotti: the rates were 22.7% in the standard fractionation arm and 33.3% in the hyperfractionation arm [49].

Machtay retrospectively performed an analysis on factors associated with severe late toxicity after concurrent pCRT in three RTOG studies, one of which was RTOG 91-11 [69]. On multivariable analysis, he found the following strong independent risk factors: older age (OR 1.05 per year), advanced T stage (OR: 3.07), larynx/hypopharynx as the primary site (OR 4.17) and neck dissection after pCRT (OR 2.39) [69]. Langendijk prospectively developed a predictive model for the most disturbing late toxic effect after p(C)RT: swallowing dysfunction. The endpoint was grade 2–4 RTOG swallowing dysfunction at 6 months (defined as: “unable to take solid food normally; swallowing semisolid food; dilatation may be indicated”). Based on the prospectively scored radiation-induced toxicity of 529 patients treated with p(C)RT, he developed a “Total Dysphagia Risk Score” (TDRS), which could be calculated by summation of the independent prognostic factors, multiplied by “risk points” (RP), which were derived from the regression coefficients from the multivariate model. A simplified table to easily read out the predicted TDRS is given in Table 8. The calculated TDRS assigns each individual patient to a risk group. In the low risk group (0–9 RP), the risk of experiencing grade 2–4 RTOG swallowing dysfunction was below 10%, and in the high-risk group (>18 RP) it was over 30%.

A hypothetical patient of the Rudat study [66] may serve as an example of a patient who will probably suffer from swallowing dysfunction: T3 hypopharynx carcinoma, 3 kg loss of weight before starting treatment, planned accelerated CRT with bilateral neck irradiation. In this constellation the calculated TDRS would amount to 31 RP, far above 18 RP, thus assigning the patient to the high-risk group with a risk of experiencing considerable swallowing dysfunction in more than 30% of cases. In contrast, a favorable but realistic example would be a T2 N0 glottic cancer patient without loss of weight who might be proposed for concomitant pCRT of the primary alone. His TRDS would amount to no more than 5 RP with pCRT as the only risk factor. Therefore, he would be assigned to the low-risk group with up to 9RP in which there was a risk of considerable swallowing dysfunction in less than 10% of patients. Langendijk´s model shows that the addition of chemotherapy to primary radiotherapy is not the main factor inducing swallowing dysfunction. Not yet included in the model is the reduction in late toxicity achieved by IMRT, which was not yet used in Rudat’s [66], Forastière’s [68], and other studies that started before 2005. Late dysphagia strongly correlates with the delivered dose to pharyngeal constrictor muscles (PCMs) [71,72,73]. In contrast to hypopharyngeal carcinoma, in glottic cancers, the inferior PCM can be spared. Thus, with modern IMRT, the incidence of swallowing dysfunction could be further reduced. There are emerging sophisticated and even more selective and perhaps more effective treatment modalities such as proton therapies in different forms [74,75,76]. However, these are still experimental and not yet ubiquitously available to date [11]. In this context, as far as effectiveness is concerned, a practical advantage of the addition of chemotherapy to pRT is that cisplatin is available in any institution treating advanced HNSCC. Its effectiveness is well proven and most radiation oncologists have abundant experience with this substance.

### 4.9. Patients’ Preferences

Even though there may be concerns of toxicity on the part of the physician, one important aspect must be taken into consideration: patients’ preferences. According to the NCCN guidelines, the choice of treatment modality depends on anticipated functional outcome, the reliability of follow-up, general medical conditions, and the patient´s wishes [14]. Functional outcomes—such as voice, swallowing, and quality of life (QoL)—are important factors when choosing a primary treatment [22]. Cancer, however, is a fundamentally life-threatening disease. This awareness becomes particularly strong, even if oneself is concerned. Blanchard conducted a systematic review of 20 studies dealing with HNSCC patient preferences, among which 9 explicitly dealt with laryngeal carcinoma. In all of the studies, being cured and surviving consistently had the highest priority. Patients would not trade large differences for functional outcomes [77]. Being cured ranked at the top of the list by 75%, and was placed in the top three by 93%, directly followed by living as long as possible by 56% [78,79]. Obviously, survival is of paramount importance, overshadowing associated toxicities and potential dysfunction [78]. Patients were more willing than nonpatients “to undergo aggressive treatments and endure acute distress in the interest of potential long-term gains (i.e., cure or longer survival)” [78]. Because of the excellent responsiveness of HPV-driven oropharyngeal cancer, there is a discussion of the deintensification of therapy. In this context, oropharyngeal cancer patients post-pCRT were asked what potential difference in cancer survival was acceptable for them to prefer pRT over pCRT. When survival rates were first presented as identical between pRT and pCRT, 90% of patients selected pRT. However, 69% switched to pCRT in the case of a reduction in the survival rate of 0% to 5% [80]. Thus, for cancer patients, survival is of paramount importance and patients would choose more aggressive treatment strategies such as pCRT instead of pRT if there was a chance to achieve higher survival rates—even at the expense of higher toxicities or possibly residual impairment.

### 4.10. For Which T2 Glottic Cancer Patients Could pCRT Be Considered?

Our data do not show a superiority of pCRT over TLM in T2 cancer patients. There is not yet statistical evidence for this assumption. Further, ideally prospective studies with larger numbers of patients are necessary to show which results can be achieved.

For the general decision between the primary conservative and the primary surgical approach, there is still no compelling evidence for the superiority of one over the other as no head-to-head comparative studies have been performed in the early stage [17]. However, the evidence thus far available from the literature and our own research suggests that TLM is superior to pRT in terms of OS and LP. Nevertheless, many criteria have to be taken into consideration for decision making, as has been shown in several valuable reviews [17,19]. Thus, intensive individual counseling is fundamental. Two prerequisites for safe endoscopic resection are optimal anatomical exposure of the tumor and sufficient expertise and caseload of the institution and the individual surgeon [81]. There should be no doubt that R0 resection is safely achievable. Otherwise, pRT (or OPL) is the better alternative as reported by Chung [47]. In his series, 90 out of 143 T1 tumors (63%) and all T2 tumors were considered not adequately resectable and thus received pRT. They report a poorer LC after TLM compared to pRT and state that this might be partly due to the inferior results of their surgical group compared to previous reports (5-year LC rate of 73.2% in T1 patients) [47]. In this case, the safer option is pRT with clearly defined standards of the radiation field. Therefore, the results are essentially dependent on the efficacy of the radiation itself. However, this may be significantly enhanced by the addition of chemotherapy and may lead to better results in T2 patients.

Until the efficacy and acceptable toxicity of pCRT are proven on a larger scale, pCRT instead of standard pRT might be offered to T2 patients with excellent compliance and the explicit wish of maximal effective treatment even at the dispense of higher toxicities. In addition, they should be in a comparably good state of health. This is important, as the addition of chemotherapy should never result in a higher incidence of treatment interruption days or even late treatment breaks, as these are associated with increased local failure rates [26]. Retrospectively, it is unclear why the early laryngeal cancer patients in our cohort study were treated with uncommon pCRT. Considering their CCI and age, colleagues 20 years ago obviously chose significantly younger and healthier patients (54.8 years compared to 63.4 and 65.3 years in the T1 surgery and pRT groups, respectively). This is in line with Machtay who reported higher risks of late toxicities in older patients (HR 1.05 per year; *p* = 0.001 [69]. However, it is in contrast to Langendijk, who reported fewer late toxicities in patients aged over 60 years compared to patients aged between 18 and 60 years (HR 1.7; 95%-CI: 1.13–2.56). Thus, the age of patients for whom pCRT should be recommended is still an open issue.

A recommendation could be to calculate the Total Dysphagia Risk Score (Table 8) to avoid increased late laryngeal toxicity rates and the patient should be assigned to the low-risk group. Pending the results of prospective randomized studies, the outcome of such pCRT-treated T2 patients could be collected for preliminary evaluation in terms of toxicity, oncological outcome and larynx preservation.

## 5. Conclusions

Twenty years ago, in Germany, the main treatment approach for laryngeal cancer over all tumor categories was surgery. Unexpectedly and in contrast to the common standards, a small number of early-stage patients received pCRT. In multivariate Cox regression analyses, the oncological outcomes in these pCRT patients were superior to pRT in terms of OS and LP. T2 patients treated by pCRT achieved better 5- and 10-year OS rates than surgically treated T1 patients. Because of the low number of patients thus far, there is no sufficient evidence of enhanced effectivity. However, the reported data of our study and additional reports from the literature show the proof of principle that by the addition of cisplatin to pCRT in T2 laryngeal cancers, results might be achieved that surpass the limits that have not yet been overcome by altered fractionation regimens. Thus, prospective randomized studies are warranted that show increased effectiveness and acceptable toxicity.

## Figures and Tables

**Figure 1 cancers-13-01601-f001:**
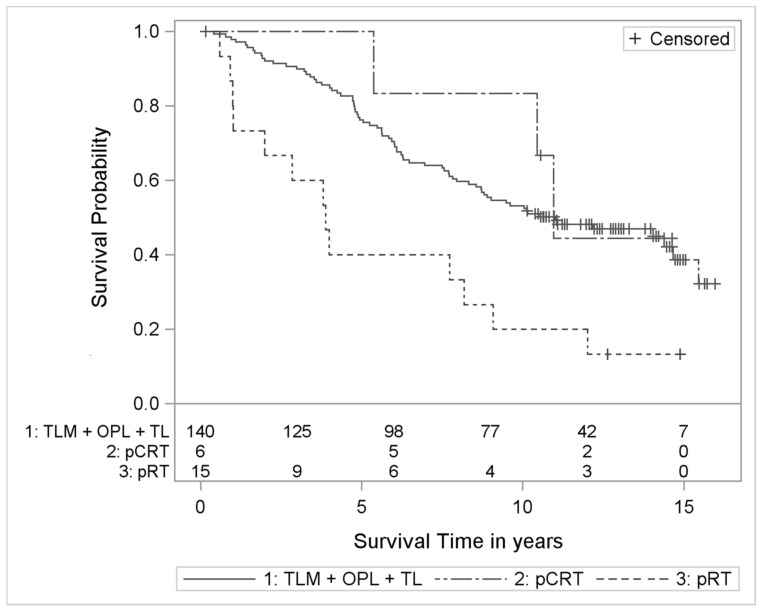
Kaplan-Meier curves with numbers at risk of T2 laryngeal cancer patients treated with pCRT or pRT compared to surgery. TLM: transoral laser microsurgery, OPL: open partial laryngectomy, TL: total laryngectomy, pCRT: primary radiochemotherapy, pRT: primary radiotherapy.

**Table 1 cancers-13-01601-t001:** Demographic and clinical characteristics of 757 laryngeal cancer patients.

Variable	Category	TLM	OPL	TL	pCRT	pRT	Total
Total	-	443	59	172	38	45	757
Age (cont) ^a^	-	62.5 (37–91)	62.1 (34–84)	61.9 (40–83)	61.4 (41–81)	64.9 (40–85)	62.4 (34–91)
Sex	Males	402 (90.7)	57 (96.6)	158 (91.9)	32 (84.2)	36 (80.0)	685 (90.5)
-	Females	41 (9.3)	2 (3.4)	14 (8.1)	6 (15.8)	9 (20.0)	72 (9.5)
CCI	0	331 (74.7)	45 (76.3)	114 (66.3)	31 (81.6)	22 (48.9)	543 (71.7)
-	1	112 (25.3)	14 (23.7)	58 (33.7)	7 (18.4)	23 (51.1)	214 (28.3)
Localization	Glottic	336 (75.8)	49 (83.1)	49 (28.5)	8 (21.1)	23 (51.1)	465 (61.4)
-	Supraglottic	96 (21.7)	7 (11.9)	57 (33.1)	20 (52.6)	14 (31.1)	194 (25.6)
-	Subglottic	4 (0.9)	0 (0.0)	8 (4.7)	1 (2.6)	1 (2.2)	14 (1.8)
-	Transglottic	4 (0.9)	0 (0.0)	38 (22.1)	6 (15.8)	3 (6.7)	51 (6.7)
-	Unknown	3 (0.7)	3 (5.1)	20 (11.6)	3 (7.9)	4 (8.9)	33 (4.4)
T-Stage	1	277 (62.5)	32 (54.2)	5 (2.9)	5 (13.2)	12 (26.7)	331 (43.7)
-	2	122 (27.5)	17 (28.8)	34 (19.8)	9 (23.7)	18 (40.0)	200 (26.4)
-	3	31 (7.0)	7 (11.9)	65 (37.8)	11 (28.9)	7 (15.6)	121 (16.0)
-	4	13 (2.9)	3 (5.1)	68 (39.5)	13 (34.2)	8 (17.8)	105 (13.9)
*n*-Stage	0	363 (81.9)	54 (91.5)	105 (61.0)	19 (50.0)	30 (66.7)	571 (75.4)
-	1	18 (4.1)	0 (0.0)	21 (12.2)	3 (7.9)	4 (8.9)	46 (6.1)
-	2	31 (7.0)	2 (3.4)	41 (23.8)	11 (28.9)	8 (17.8)	93 (12.3)
-	3	1 (0.2)	0 (0.0)	1 (0.6)	3 (7.9)	2 (4.4)	7 (0.9)
-	X	30 (6.8)	3 (5.1)	4 (2.3)	2 (5.3)	1 (2.2)	40 (5.3)
UICC stage	I	265 (59.8)	31 (52.5)	3 (1.7)	3 (7.9)	10 (22.2)	312 (41.2)
-	II	98 (22.1)	17 (28.8)	25 (14.5)	6 (15.8)	15 (33.3)	161 (21.3)
-	III	39 (8.8)	6 (10.2)	57 (33.1)	10 (26.3)	6 (13.3)	118 (15.6)
-	IV	41 (9.3)	5 (8.5)	87 (50.6)	19 (50.0)	14 (31.1)	166 (21.9)
Adj. treat.	None *	360 (81.3)	52 (88.1)	93 (54.1)	38 (100)	45 (100)	588 (77.7)
-	aRT	74 (16.7)	7 (11.9)	62 (36.0)	0 (0.0)	0 (0.0)	143 (18.9)
-	aCRT	5 (1.1)	0 (0.0)	17 (9.9)	0 (0.0)	0 (0.0)	22 (2.9)
-	aCT	4 (0.9)	0 (0.0)	0 (0.0)	0 (0.0)	0 (0.0)	4 (0.5)

TLM: transoral laser microsurgery, OPL: open partial laryngectomy, TL: total laryngectomy, pCRT: primary chemoradiotherapy, pRT: primary radiotherapy, CCI: Charlson Comorbidity Index, ^a^ Age (cont): age (continuous) in years: mean with minimal and maximal ages, Localization: primary tumor localization; Adj. treat.: adjuvant treatment; * in some charts the entry was missing, i.e., no adjuvant treatment or unknown status, aRT: adjuvant radiotherapy, aCRT: adjuvant chemoradiotherapy, aCR: adjuvant chemotherapy.

**Table 2 cancers-13-01601-t002:** Five- and 10-year DSS and OS of all laryngeal cancer patients of the cohort over T categories and treatment.

T Stage, Therapy (*n*, DSS/OS *)	5-Year DSS [%] (95%-CI)	10-Year DSS [%] (95%-CI)	5-Year OS [%] (95%-CI)	10-Year OS [%] (95%-CI)
T1	-	-	-	-
OP (307/314 *)	95 (92–97)	93 (89–95)	80 (75–84)	59 (53–64)
pCRT (5/5)	100 (100–100)	100 (100–100)	100 (100–100)	80 (20–97)
pRT (12/12)	100 (100–100)	75 (31–93)	67 (34–86)	42 (15–67)
T2	-	-	-	-
OP (172/173 *)	82 (76–88)	73 (65–79)	69 (62–76)	48 (40–55)
pCRT (9/9)	100 (100–100)	100 (100–100)	89 (43–98)	67 (28–88)
pRT (18/18)	53 (28–73)	53 (28–73)	33 (14–55)	17 (4–37)
T3	-	-	-	-
OP (99/103 *)	76 (65–83)	65 (53–74)	61 (50–69)	35 (26–45)
pCRT (11/10)	70 (33–89)	26 (1–66)	55 (23–78)	9 (1–33)
pRT (5/7 *)	20 (1–58)	20 (1–58)	14 (1–46)	14 (1–46)
T4	-	-	-	-
OP (79/84 *)	52 (39–63)	40 (27–52)	38 (28–48)	20 (12–29)
pCRT (13/13)	8 (0–29)	8 (0–29)	8 (0–29)	8 (0–29)
pRT (8/8)	13 (1–42)	13 (1–42)	13 (1–42)	13 (1–42)

* Differences in numbers of patients between OS and DSS due to unknown causes of death. DSS: disease-specific survival, OS: overall survival, OP: surgery, pCRT: primary chemoradiotherapy, pRT: primary radiotherapy.

**Table 3 cancers-13-01601-t003:** Multivariable Cox regression analysis.

Variable	Category	*p*-Value	HR	95%-CI	
Age	10 years units	<0.0001	**1.573**	**1.368**	**1.810**
Sex	Female vs. male	0.0670	0.665	0.429	1.029
CCI	1 vs. 0	0.0006	**1.564**	**1.212**	**2.018**
Therapy	pCRT	0.3685	0.704	0.327	1.514
-	pRT	0.0094	**1.793**	**1.154**	**2.785**
T category	T2 vs. T1	0.0675	** **1.278** **	** **0.982** **	** **1.662** **
Localization	supraglottic	<0.0001	**1.925**	**1.432**	**2.588**
-	subglottic	0.3721	1.690	0.534	5.348
-	transglottic	0.0921	2.070	0.888	4.825
-	unknown	0.4939	1.368	0.558	3.354

**Bolded black**: statistically significant, **bolded grey**: strong tendency without reaching the level of significance; HR: hazard ratio, 95%-CI: 95%-confidence interval, CCI: Charlson Comorbidity Index, pCRT: primary chemotherapy, pRT: primary radiotherapy.

**Table 4 cancers-13-01601-t004:** Survival and larynx preservation after TLM in T1, T2 and T1 + T2 laryngeal cancer patients of all primary tumor sites.

Variable	T Category	Patients *n* (%)	5-Year [%] (95%-CI)	10-Year [%] (95%-CI)
DSS	T1	270 * (62.6)	96 (92–98)	93 (89–96)
-	T2	122 * (28.3)	85 (77–90)	78 (68–84)
-	T1 + T2	392 * (91.0)	92 (89–95)	88 (84–91)
OS	T1	277 (62.5)	80 (75–84)	58 (52–63)
-	T2	122 (27.5)	73 (64–80)	50 (41–59)
-	T1 + T2	399 (90.1)	78 (73–82)	55 (50–60)
LP	T1	277 (62.5)	93 (89–96)	93 (89–96)
-	T2	122 (27.5)	82 (73–88)	82 (73–88)
-	T1 + T2	399 (90.1)	90 (86–93)	90 (86–92)
LFS	T1	277 (62.5)	74 (68–79)	54 (48–60)
-	T2	122 (27.5)	65 (56–73)	46 (36–54)
-	T1 + 2	399 (90.1)	71 (67–76)	52 (47.57)

DSS: disease-specific survival, OS: overall survival, LP: larynx preservation: event defined as occurrence of total laryngectomy, LFS: laryngectomy-free survival: OS with functional larynx, event defined as occurrence of either laryngectomy or death. * Differences in numbers of patients between OS and DSS due to unknown causes of death.

**Table 5 cancers-13-01601-t005:** Comparison of outcomes between TLM and pRT in early-stage glottic cancers.

Meta-Analysis T Categories Patients (*n*)/Studies (*n*)	LC [HR (95-CI)]	DSS [HR (95%-CI)]	OS [HR (95%-CI)]	LP [HR (95%-CI)]
Higgins, 2009 [41], Tis-T2 7676/26	0.81 (0.51–1.35)	n.a.	**1.48** **(1.19**–**1.85)**	TL-free survial 0.73 0.39–1.35 *^2^
Mo, 2017 [18], T1 1238-1452/11	0.98 (0.7–1.38)	n.a.	**1.35** **(1.02**–**1.79)**	**5.81** **(3.36**–**10.05)**
Guimaraes, 2018 [19] *^1^ Tis-T1a 1034-1481/6-10	WMD TLM vs. pRT −0.01 [−0.07, 0.04] Z = 0.45; *p* = 0.65	WMD TLM vs. pRT −0.02 [−0.04, −0.00] Z = 2.03; *p* = 0.04	**WMD** −**0.05 [**−**0.09,** −**0.00)** **Z = 1.97; *p* = 0.05**	**WMD** −**0.10 [**−**0.13,** −**0.07]** **Z = 6.53; *p* < 0.00001**
Ding, 2019 [20], T1-T2 2480/18	1.19 (0.76–1.85)	1.60 (0.89–2.88)	**1.39** **1.06**–**1.81)**	**3.85** **1.92**–**7.72**
Vaculik, 2019 [21], T1 1987/17	1.19 0.79–1.81	2.70 1.32–5.54	**1.52** **1.07**–**2.14**	**6.31** **3.77**–**10.56**

TLM: transoral laser microsurgery, pRT: primary radiotherapy, LC: local control, DSS: disease-specific survival, OS: overall survival, LP: larynx preservation, HR hazard ratio, 95%-CI: 95%-confidence interval. **Bolded** values were statistically significant. *^1^ Guimaraes used the weighted means difference (WMD). The other authors expressed their results in hazard ratios (HR); *^2^ Laryngectomy-free survival data in Higgins meta-analysis were based on 4 among the 26 studies in which the Spector study [42] had a weight of 51.8%. This study, however, was not representative for TLM studies in general. It was published in 1999 and reports rather preliminary results from the built-up phase of laser surgery. Six patients among the 61 TLM patients (9.8%) and 29/194 pRT patients (14.9%) underwent total laryngectomy.

**Table 6 cancers-13-01601-t006:** Oncological outcome after pRT in T1 and T2 glottic cancer patients.

Study Stage (*n* pts.)	5-Year-LC [%]	5-Year DSS [%]	5-Year OS [%]	5-Year-LP [%]
Warde, 1998 [23]	-	-	-	-
T1a (403)/T1b (46)	91/82	n.a.	75.8 (T1+T2)	n.a.
T2 (286)	69	n.a.	-	n.a.
Frata/Cellai 2005 [24,25]	-	-	-	-
T1a (660)/T1b (171)	84/81	95	77	n.a.
T2 (256)	73	86	59	-
Groome, 2006 [26]	-	-	-	-
T1 (491)	82	93	77	n.a.
T2 (213)	63	81	70	n.a.
Chera, 2010 [27]	-	-	-	-
T1a (253)/T1b (72)	94/93	97/99	82/83	n.a.
T2a (163)/T2b (95)	80/70	94/90	76/78	n.a.
Tong, 2012 [28]	-	-	-	-
T1a (324) /T1b (109)	92/89	98	89	87 (T1+2)
T2 (262)	79	98	89	-
Al-Mamgani, 2013 [29]	-	-	-	-
T1 (719)	92	n.a.	n.a.	n.a.
T2 (331)	78	n.a.	n.a.	n.a.

LC: local control, DSS: disease-specific survival, OS: overall survival, LP: larynx preservation. Values specified according to a and b subcategories as far as presented in the publications, *n* pts.: number of patients, n.a.: not available.

**Table 7 cancers-13-01601-t007:** Oncological outcome after pRT and pCRT in T2 glottic cancer patients in three series of Furusaka.

Study Treatment (*n* pts.)	5/10-Year OS [%]	5/10-Year-LP [%]
Furusaka, 2012 [62]	-	-
pRT alone (57)	88.5/73.5	60.4/50.1
Furusaka, 2012 [61]	-	-
pCRT (Carbo AUC 1.5) (25)	83.4/77.0	79.0/73.0
pCRT (Carbo AUC 2.0) (25)	95.7/91.1	79.0/73.0
Furusaka, 2013 [63]	-	-
pCRT (Cis/5-FU) (32)	95.3/95.3	75.1/75.1

*n* pts.: number of patients; 5/10-year -OS: 5- and 10-year overall survival, LP: larynx preservation, pRT: primary radiotherapy, pCRT: primary radiochemotherapy, Carbo: carboplatin; AUC: area under the blood concentration-time curve; Cis: cisplatin; 5-FU: 5-fluorouracil.

**Table 8 cancers-13-01601-t008:** Calculation of the Total Dysphagia Risk Score (TDRS): Sum of the respective risk points (RP) per category; subsequently, assignment to the respective risk groups. RP of hypopharynx as primary site extrapolated to 5 from HRs in univariate analysis with larynx set to 1: nasopharynx: HR 7.74 (≥9 RP) oropharynx: HR 5.16 (≥7 RP) and thus hypopharynx: HR 3.48 (≥5 RP); Interm. risk: intermediate risk; modified from Langendijk et. al., 2009 [70].

Category	Variable	Risk Points			
T classifaction	T1–T2	0			
	T3–T4	4			
Neck irradiation	Primary alone ± ipsilateral	0			
	Primary ± bilateral	9			
Weight loss	none	0			
	1–10%	5			
	>10%	7			
Primary tumor site	Larynx	0			
	Hypopharynx	5			
	Oropharynx	7			
	Nasopharynx	9			
Treatment modality	Conventional RT	0			
	Accelerated RT	6			
	Concomitant CRT	5			
					-
	**Sum RP = TDRS:**		0–9 RP	10–18 RP	>18 RP
			Low risk	Interm. risk	High risk
			≤10%	10–30%	>30%
		**Individual risk group:**			

## Data Availability

The datasets generated and analyzed during the current study are available from the corresponding author on reasonable request.

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
