# Peer review of "Could Primary Chemoradiotherapy in T2 Glottic Cancers Yield Results Comparable to Primary Radiotherapy in T1? Considerations from 531 German Early Stage Patients"

_cancers, 2021, doi:10.3390/cancers13071601_

Round 1
Reviewer 1 Report
Thank you for very much for this well written article ( with very extensive literature), that can pave the way for bigger studies in the near future. Although already mentioned in the study, i am still curious over the possible reasons for giving pCRT for T1/T2 lesions. Was this brought about by failure rates/patterns in the past?
For the T2 lesions, what was the failure pattern after OP/pRT? Was it primarily locoregional/distant?
Author Response
Answer Reviewer 1
Dear Reviewer,
Thank you friendly for your very kind and favorable review!
Here our answers to your questions in detail:
@1 - Although already mentioned in the study, I am still curious over the possible reasons for giving pCRT for T1/T2 lesions. Was this brought about by failure rates/patterns in the past?
Answer: We very much appreciate your interest in possible reasons for giving pCRT in T1/2 lesions. Consequently, we have checked in which of the five institutions and in which years the patients were recruited for pCRT. All T1 and a part of the T2 patients were only sporadically indicated for pCRT in all five institutions over the whole recruitment period. Two of the institutions indicated pCRT in only one single patient. We found, however, a series of five T2 patients in one of the centers in the years 1998-2000. Indeed, there might have been some recurrences in patients treated by pRT only just before the recruitment period started so that under this impression there was the wish for a more effective therapy. But as we have only the data of the patients recruited in the period starting from 1998 this explanation is speculative. Therefore, we would not discuss it in the paper.
@2 - For the T2 lesions, what was the failure pattern after OP/pRT? Was it primarily locoregional/distant?
Answer: In the whole cohort only in 5 patients, distant metastases have been verified. But these were in supraglottic tumors of higher stages. There were some metachronous second primaries as cause of death decreasing the OS rate. But diminishment of DSS rates in OP and pRT patients was caused by locoregional recurrences.
We hope that our answers meet your expectations.
With kind regards
Gerhard Dyckhoff
Reviewer 2 Report
This manuscript is well-written, thorough and should be of high interest to readers. I noted the following minor concerns.
- The title is very long and should be modified to include the fact that the cases were all from German.
- line 142 should indicate "table 2" not "tables 2"
- I am confused that the Discussion begins on line 143 and then Tables 5-9 are presented. Shouldn't they be in the results section?
- The sentence describing the example that begins on line 494 needs to be edited for clarity and grammar.
Author Response
Dear Reviewer,
Thank you friendly for your very favorable review!
We have implemented the corrections as follows:
@1 - The title is very long and should be modified to include the fact that the cases were all from German.
Answer: We have specified that patients were from Germany. “Cancers” might sound smoother than “carcinoma”. We have shortened the second part of the title as follows:
“Could Primary Chemoradiotherapy in T2 Glottic Cancers Yield Results Comparable to Primary Radiotherapy in T1? Considerations from 531 German Early Stage Patients.”
@2 – 1. line 142 should indicate "table 2" not "tables 2"
Answer: “Table 2” is now correct
@3 - I am confused that the Discussion begins on line 143 and then Tables 5-9 are presented. Shouldn't they be in the results section?
Answer: You are right that the Tables 5-9 are the results of our literature search. But they are part of the Discussion where we have implemented them in the respective section as summary and illustration of what is written in the text. For the reader, when reading the discussion, they might be easier to find when they are inserted nearby. Here we followed the instruction given in the MDPI template: “Tables should be placed in the main text near to the first time they are cited.”
@4 - The sentence describing the example that begins on line 494 needs to be edited for clarity and grammar.
Answer: The sentence has been changed as follows: “A hypothetical patient of the Rudat study may serve as an example of a patient who will probably suffer from swallowing dysfunction: T3 hypopharynx carcinoma, 3 kg loss of weight before starting treatment, planned accelerated CRT with bilateral neck irradiation. In this constellation the calculated TDRS would amount to 31 RP (…)”
We hope that our changes and corrections meet your expectations.
With kind regards
Gerhard Dyckhoff